



# High accuracy calculation and data quality evaluation of
# ship emissions based on the sniffer method
Letian Zhu[1,2], Fan Zhou[1,2]
[1]College of Information Engineering, Shanghai Maritime University, Shanghai 201306, China
[2]Shanghai Engineering Research Center of Ship Exhaust Intelligent Monitoring, Shanghai 201306, China
*Correspondence to*: Fan Zhou (fanzhou_cv@163.com)
**Abstract.** More attention has been paid to the air pollution caused by ship emissions; hence the
establishment of accurate emission inventories is an important means to assess the impact on the
environment and human beings. The emission factor is an important parameter in the process of
compiling the ship emission inventory, yet there is some uncertainty in its estimation based on the sniffer
method. In this study, taking the calculation of $SO_2$ emission factors as an example and aiming at the
selection of gas measurement values using the sniffer method, the concept of standard deviation of peak
density was proposed to determine the optimal integral interval length of the measured values of $SO_2$ and
$CO_2$. Then, the improved Manhattan distance was used to characterize the position of the peak points in
the $SO_2$ and $CO_2$ average series. Using the dynamic time warping algorithm, the corresponding
relationship of the peak points in the average series of the measured gases was determined, and the global
optimal peak points were selected from it. To evaluate the credibility of calculated emission factors, 16
evaluation indexes that reflect the characteristics of the measured data were selected. The confidence
interval of 95% of each evaluation index was calculated using self-development sampling of the
measured data, and the evaluation result of the evaluation index for the quality of the measured data was
obtained. Combined with the data quality label, the indexes with high correct rate were screened. Finally,
the evaluation scores were determined according to these selected indexes. We collected a total of 148
sets of "$SO_2$+$CO_2$" measurement data between 2019 and 2021 using the unmanned aerial vehicle sniffing
monitoring system in the Waigaoqiao Port area of Shanghai, China for verification using the method
proposed in this study. The results show that for this data set, 12 s is the most suitable integral length,
with which the algorithm can automatically calculate the emission factor. The screening results of the
global optimal peak points of 129 groups of data are consistent with those of artificial screening, with a
correct rate of 87.16%. The accuracy of the combined evaluation of sample entropy ($SO_2$), information
entropy ($SO_2$), skewness ($CO_2$) and quartile spacing ($SO_2$) is 71%. Previous calculation of the emission



factor of ships mainly focused on different conditions such as time, region, fuel, engine, ship type, and
navigation status. Our in-depth study proposes a high accuracy ship emission factors calculation method
and an evaluation of the quality of the measurement data that reduces uncertainty in the current sniffer
technique monitoring ship emission research.
**1 Introduction**
In the past decade, the development of the global shipping industry has accelerated (UNCTAD, 2021),
resulting in increasingly serious emission problems (Chen et al., 2019). Gases and particles emitted by
ships will not only pollute the natural environment, but also have an impact on human health (Liu et al.,
2016). $SO_2$ causes frequent acid rain damage to the environment (Matthias et al., 2010) such as the
erosion caused to 50% and 30% of the forests in Germany and Poland and Switzerland, respectively
(Mohajan et al., 2018). Fine particulate matter ($PM_{2.5}$) can cause lung cancer and other heart and lung
diseases, and results in 2.2 million to 3.3 million deaths worldwide each year (Sofiev et al., 2018). In
2015, a total of $20.1 \times 10^6$ tons of $NO_x$, $11.5 \times 10^6$ tons of $SO_x$, and $1.54 \times 10^6$ tons of PM were emitted by
global shipping operations (Sofiev et al., 2018). In 2017, EU shipping emissions generated $2.6 \times 10^6$ tons
of $SO_2$ and $7.7 \times 10^6$ tons of $NO_2$ (Jonson et al., 2020).
Accurate ship emission inventory is not only the baseline data for analyzing the law of ship emissions,
but also the scientific basis for controlling and optimizing supervision measures in relevant management
areas (Zhang et al., 2017; Zhang et al., 2017). The calculation of ship emission inventory is carried out
using two approaches: top-down and bottom-up. The top-down method is based on the fuel consumption
of the ship without considering its specific location, and is suitable for the calculation of the long-term
source list on the global scale. Kesgin et al. (2001) used the top-down method to calculate the emissions
of CO, $CO_2$, $PM_{2.5}$ from ships in the Turkish Strait. Corbett et al. (1997) studied the global emission
inventory using ship fuel. The bottom-up method directly estimates emissions based on the motion state
and attributes of the ship, which is more accurate than the top-down method. In recent years, due to the
rapid development of the Automatic Identification System (AIS), it is more convenient to obtain the real-
time operation status of ships, hence this method is widely used in the research of emission inventory.
Papaefthimiou et al. (2016) used a bottom-up approach based on port ship activity to calculate the $NO_x$,
$SO_2$, $PM_{2.5}$ emitted by international cruise ships in 18 ports in Greece. Tichavska et al. (2015) obtained



ship activity data through the AIS and studied the monthly relationship between ship emissions and vessel
type in Las Palmas Port.
Parameter information such as ship speed and position, main engine power, auxiliary power, and
emission factors are needed when establishing an emission list. Among them, the static and dynamic
information of the ship can be obtained directly using AIS data and other channels, whereas the emission
factor can only be determined by measurement, thus the accuracy of the measurement directly determines
the accuracy of the emission list (Ekmekçioğlu et al., 2020; Yang et al., 2021; Toscano et al., 2021).
There are two ways to measure emission factors: fuel- and power-based. The former can be calculated
by measuring the concentration of $CO_2$ and other pollutants, while the latter needs to obtain real-time
data of mainframe power, auxiliary power, and operation mode. The two measurements can be converted
to each other when the ship fuel consumption rate is known (Zhang et al., 2016). Sniffer technique is one
of the methods based on the fuel emission factor measurement, which can quickly and accurately
calculate the ship plume. Balzani Lööv et al. (2014) used the sniffer technique among others to compare
the measures of the emission factor and fuel oil sulfur content of ships in the port of Rotterdam. The
mobile sniffer technique was the most convenient and accurate, with an average random error of 6% for
$SO_2$ emission factors. Beecken et al. (2014) used a small aircraft carrying sniffer equipment to monitor
exhaust emissions from 158 ships in the Baltic Sea and North Sea. The average emission factor of $SO_2$
was $18.8 \pm 6.5 kg_{fuel}^{-1}$, and approximately 85% of the monitored ships met the sulfur content limit set
by the International Maritime Organization. Many studies have shown that the emission factor and its
accuracy is an important parameter in the compilation of ship emission inventory (Moreno-Gutiérrez et
al., 2015; Zhang et al., 2017; Ekmekçioğlu et al., 2020). Therefore, many emission factors measurements
have been proposed to obtain accurate results. These experiments focused more on measuring and
obtaining emission factors under different conditions such as time, region, fuel, engine, ship type, and
navigation status (Sinha et al., 2003; Cooper, 2003; Burgard and Bria, 2016; Peng et al., 2016; Betha et
al., 2017; Liu et al., 2018; Bai et al., 2020). However, there is still a lack of in-depth research on how to
accurately calculate the emission factor from measurement data and how to evaluate the quality of the
measured data.
Taking $SO_2$ as an example, the principle of the sniffer technique is based on the 85% and 87% stability
of the carbon content in ship fuel. The concentration ratio of $CO_2$ to $SO_2$ generated by fuel combustion
is equal to the molar ratio of carbon to sulfur in the fuel and will not change due to tail gas dilution, thus



the measurement of $CO_2$ and $SO_2$ concentration can be used to calculate the emission factor (Huang et
al., 2021). However, various uncertain factors in the calculation process cause nonnegligible interference
to the emission factor, which mainly exists in three aspects: first, because the response time of different
gas sensors varies, it is difficult to select the measured value of gas at the same time as the calculated
emission factor. Therefore, the general method is to select the gas measurement values for a period for
the cumulative calculation to reduce the error caused by the response time of different gas sensors that
cannot be synchronized. Zhou et al. (2020) proposed to regard the accumulation process as an integral,
and then divide the result by the time interval, so as to convert the gas measurement into an average value
to find a method that selects the global optimal peak point of the gas average value. The problem of
selecting the appropriate integral interval is selecting the global peak point in the average set of measured
data. However, the time interval of the integral interval mainly depends on experience. Second, based on
the first problem, the selection of the global optimal peak point directly determines the accuracy of the
emission factor calculation. However, multiple peaks of $SO_2$ and $CO_2$ will appear in gas measurements
over a period. Therefore, there is a lack of in-depth research on how to establish a matching relationship
between the peak points to determine the global optimal peaks of $SO_2$ and $CO_2$. Third, various
environmental and equipment factors in the process of ship exhaust gas measurement will interfere with
the gas measurement value, hence there is no objective evaluation method for the gas measurement value
at present.
This study makes an in-depth study and analysis of the above three problems. Aiming at the first problem,
the concept of peak density standard deviation is proposed to measure the ability of an integral interval
to represent the changing data. The larger the value of the standard deviation of peak density, the more
obvious the changes of the data processed by the corresponding integral interval, the clearer the peak
trend, and the more accurate the selection of the global optimal peak point. Aiming at the second problem,
to select the optimal peak point from many peak points, it is necessary to establish a matching relationship
between the average series of the two gases and judge the rationality of all the matching relations. The
dynamic time warping (DTW) algorithm was proposed by Itakura (Itakura, 1975), and its function is to
measure the similarity of two time series. Dmytrów et al. (2021) recently used the DTW algorithm to
evaluate the similarity between energy commodity prices and the daily cases time series for the
coronavirus disease (COVID-19). Li et al. (2020) used the DTW algorithm to compare the corresponding
relationship between ship trajectory time series, which improved the performance of navigation trajectory



modeling. Applying the DTW algorithm to the screening of peak points can obtain the similarity between
$CO_2$ and $SO_2$ sequences and the matching relationship between the average sequences of their peak points.
Further, combining the criteria for selecting the optimal peak points proposed by Zhou et al. (2019) can
select the appropriate gas measurement value for the calculation of emission factors. Aiming at the third
problem, 16 evaluation indexes that reflect the data quality are put forward, in which the evaluation index
for a single gas measurement value is divided into two cases to use the $SO_2$ and $CO_2$ measurement values
for evaluation, respectively. The 95% confidence interval of the evaluation indexes are calculated by
self-developing sampling for a number of measured data, and the evaluation results of these indexes for
the quality of the measured data are given accordingly. Combined with the quality label of the measured
data, the correct rate of the index for data quality evaluation can be obtained, and the evaluation indexes
with higher accuracy can be selected to form a set for joint evaluation, so as to further improve the
accuracy of the gas measurement data evaluation. Finally, using the method proposed in this study, the
ship exhaust data measured in Shanghai Waigaoqiao Port are tested, and it is verified that this method
can find the appropriate global optimal peak point with high accuracy and then calculate the emission
factor, using indexes to jointly evaluate the quality of the measured data.
**2 Theory**
**2.1 Method of calculating ship emission factors**
The sniffer method is based on three assumptions: first, the carbon content in different ship fuels is
approximately 87% similar. Second, the combustion of carbon and sulfur in ship fuel produces almost
all $CO_2$ and $SO_2$, while the rest of sulfur and carbon oxides account for only a small part, which can thus
be ignored. Third, when the tail gas generated by marine fuel combustion is diluted in the air, the ratio
of $CO_2$ to $SO_2$ does not change (Hu et al., 2018). Accordingly, the $SO_2$ emission factor can be calculated
by measuring the gas concentration of $SO_2$ and $CO_2$ over a period and accumulating the ratio respectively.
The formula is (Beecken et al., 2014):
$$EF_{SO_2}[g\ kg_{fuel}^{-1}] = \frac{m(SO_2)}{m(fuel)} = \frac{M(SO_2)\cdot\Sigma[SO_{2,ppb}]}{M(C)/0.87\cdot\Sigma[CO_{2,ppm}]} = 4.64\frac{\Sigma[SO_{2,ppb}]}{\Sigma[CO_{2,ppm}]} \tag{1}$$

where $m(\cdot)$ is the mass, $M(C)$ is the relative atomic mass of carbon, $M(SO_2)$ is the relative molecular
mass of sulfur dioxide, and $\Sigma[\cdot]$ is the cumulative summation of the concentration of the gas to be
measured over time.



Considering that the sensor cannot achieve complete synchronization, the method of accumulating the
measured values over a period can be adopted instead of using one measured value, that is, the calculation
of the emission factor can be more stable by integrating the measured values of gas over a period. In this
study, we calculate the $SO_2$ emission factor by converting gas measurements into average values, as
shown in Eq. (2) (Zhou et al., 2020):
$$EF_{SO_2}\left[g\ kg_{fuel}^{-1}\right] = 4.64\frac{\int(SO_{2,peak}-SO_{2,bkg})dt[ppb]\big/t}{\int(CO_{2,peak}-CO_{2,bkg})dt[ppm]\big/t} = 4.64\frac{AVG(SO_{2,peak})-AVG(SO_{2,bkg})}{AVG(CO_{2,peak})-AVG(CO_{2,bkg})} \qquad (2)$$
where $SO_{2,peak}$ is the peak of $SO_2$ in the measured data, $SO_{2,bkg}$ is the background value of $SO_2$ in the
measured data, $t$ is the length of the integral interval, $\int(\cdot)dt$ is the integral calculation function of the
integral interval of t seconds, and $AVG(\cdot)$ is the function of calculating the average measured value in t
seconds.
Using the above transformation, the problem of selecting the integral interval of gas using the sniffer
technique is selecting the integral interval and peak point. The empirical value of 10 s was used by Zhou
et al. (2020) in the integration interval, and then by observing and analyzing the changing trend of the
peak point in the average series, the appropriate global optimal peak point was selected for the calculation
of emission factors. Through the above methods, more accurate results can be calculated. However, the
length of the selected integral interval belongs to empirical value, which lacks theoretical demonstration.
In addition, when there are many peak points in a period, there is also uncertainty about how to pair $SO_2$
and $CO_2$.
**2.2 Dynamic time warping algorithm**
In order to eliminate the inconsistency of response time between $SO_2$ and $CO_2$ sensors, it is necessary to
find the matching relationship between the peak points on the average sequence of $SO_2$ and $CO_2$, so that
the global optimal peaks with corresponding relationship can be screened. According to the distance
relationship between peak points, this study uses the DTW algorithm to establish the matching
relationship between the $SO_2$ and $CO_2$ peak points. The purpose of the DTW algorithm is to find the
difference between each data point in the target and standard time series, calculate the minimum value
after the difference accumulation, and determine the corresponding path, which is used to match the peak
points of $SO_2$ and $CO_2$ in this study. The target time series is marked as $X = (x_1, x_2, \dots, x_n)$. The standard
time series is written as $Y = (y_1, y_2, \dots, y_m)$. In this study, we correspond to the results of peak point



extraction from the $SO_2$ and $CO_2$ average series. $f$ represents the distance between the points on the
target time series and its corresponding standard time series in an unbiased ideal state:
$$d(i,j) = f(x_i, y_j) \tag{3}$$
The distance matrix D is obtained by calculating the distance between the data points and their
corresponding points in X and Y:
$$D_{ij} = d(i,j) \tag{4}$$
The shortest distance is to use the cost matrix $D_{cost}$ iteration to calculate the dynamic programing path
distance between the target and the standard sequences, and the shortest distance path represents all the
matching relations of the data points on the two sequences.
$$D_{cost}(i,j) = D(i,j) + \min\left(D_{cost}(i-1), D_{cost}(j-1), D_{cost}(i-1, j-1)\right) \tag{5}$$
The path corresponding to the shortest distance matrix between the $CO_2$ and $SO_2$ peak points is obtained
by the DTW algorithm, that is, the matching relationship between the peak points in the $SO_2$ and $CO_2$
average sequence is obtained, which can correct the deviation sequence.
**3 Method**
The high accuracy calculation of emission factors needs to select the measured values of stable air flow
over a period and eliminate the influence of various uncertain factors as much as possible. In this study,
we first determine the selection method of the optimal integral interval length. Then, use the DTW
algorithm to find the matching relationship between the peak points on the $CO_2$ and $SO_2$ average series,
and select the global optimal peak points. Finally, a quality evaluation method of ship exhaust
measurement data is proposed, which is used to evaluate the reliability of the measurement results. The
specific process is divided into the following four steps:
1.    Selection of integral interval length and extraction of peak points. The concept of standard
deviation of peak density is put forward to analyze the distribution law and changing trend of peak points
in the measured value sequence, the larger the value of standard deviation of peak density is, the longer
the corresponding integral interval can make the calculation of emission factors more stable. The
measured value is converted to the average value of the length of the integral interval, and all the peak
points in the average sequence are extracted;
2.    DTW-based matching. The DTW algorithm based on the improved Manhattan distance is used





to correct the average sequence and obtain the matching relationship between $SO_2$ and $CO_2$ peak points;

3.     Filtering matching relationships. The matching relationship between the peak points in some

measured data is established, and all the matching relations are taken as samples for k-means mean
clustering. If they can be stably divided into two categories, it shows that the dividing line between the
two categories can distinguish between normal and abnormal changes, thus the threshold of concentration
change is found. Combined with the time span threshold, the results from step 2 are screened and the
abnormal matching results are eliminated. The matching result containing the maximum average value
of $SO_2$ is found among the reserved matching results, and the matching result containing the maximum
average value of $CO_2$ is selected as the global optimal peak point from all the matching results found;

4.     Evaluation of measurement data quality. Sixteen evaluation indexes, which can characterize

the data quality are proposed. For several measurement data obtained, the 95% confidence interval of
each index is calculated by self-expansion sampling. If the index value is in the confidence interval, it is
marked as 1, otherwise 0. Combined with the quality label of the measured data, some indexes with
strong representation ability are selected from the 16 indexes to form a set of indexes for joint evaluation.
The distance between the index value of the measured data and overall mean of the central position of
the confidence interval is calculated, and the ratio of the distance to the unilateral length of the confidence
interval is obtained (a ratio greater than 1 is reassigned to 1). In joint evaluation, if the calculated ratio of
all indexes is 1, the quality of the measured data is judged to be poor; otherwise, the average value of all
ratios less than 1 is calculated, and the closer the mean is to 0, the better the quality of the measured data
is.
In Section 3.1, the principle and method of selecting the length of integral interval are introduced in detail,
and the definition of peak point is explained. Section 3.2 introduces the use of the DTW algorithm to
match peak points so as to correct the average sequence of $SO_2$ and $CO_2$. At the same time, a distance
calculation method based on the Manhattan distance is proposed, which can represent the distance
relationship between the peak points in the average sequence. The calculation method of the threshold
used to eliminate abnormal matching relations is described in detail in Section 3.3. Section 3.4 describes
the specific method of selecting indexes from the 16 evaluation indexes to form an index set for joint
evaluation.





### 3.1 Selection of integral interval length and extraction of peak points

When sniffer equipment is used to monitor the ship exhaust, because the response time of $SO_2$ and $CO_2$ sensors cannot be completely synchronized, there is a deviation between the time series of $CO_2$ and $SO_2$ measurements. Using the appropriate integral interval length (set to t) to convert the measured value per second into the average value within t seconds can reduce the deviation between time series to a certain extent.

However, setting t often adopts an empirical value, which is not supported by theoretical basis generating great uncertainty. If the selected t value is too small, there will be too many small peaks in the overall measurement data set, or a peak composed of multiple data points, which causes great difficulties in the selection of peak values, resulting in great instability in the calculation of emission factors. On the other hand, if the selected t value is too large, the fluctuation of the whole data set is relatively smooth, which cannot show a representative peak trend, which will hinder the selection of the peak value and have a great impact on the calculation of emission factors.

This study proposes a method to determine the optimal integral interval from the point of view of data mining, to reduce the uncertainty of artificial selection of empirical values. To select a suitable integral interval for preprocessing, it is necessary to analyze multiple alternative intervals, because the peak points need to be composed of at least three data points, and very large intervals will excessively smooth the data change trend. Therefore, we only need to find the corresponding relationship between the data change trend and interval, thus the selection range of the integral interval length is from 3 s to 30 s. The sliding window algorithm is used to traverse the measured values, with the window size representing the length of the alternative integral interval; the window moves one point at a time, and the ratio of the peak points of each window to the total data points of the window is calculated. The ratio is the peak density of the window, and the peak density standard deviation of the length of the alternative integral interval is calculated to reflect its changing trend in the measured value. Comparing the peak density standard deviation of each alternative integral interval length, the larger the value, the more the length of the integral interval can reflect the fluctuation of the peak density in the measured value. The specific steps of this method are as follows:

1.  Assuming that there are n measurements in a complete measurement process, the sliding window algorithm is used to traverse all the measurements. The window size is the length of the selected integral interval, and the window moves one measured value each time;





2. If a measurement point is larger than the left and right adjacent measurement points, the point is defined as the peak point. The number of peak points in each window is calculated, and the peak density of the window is calculated following Eq. (6).

$$density(j) = \frac{count_{peak(j)}}{count_{data(j)}} \tag{6}$$

where $density(j)$ represents the peak density of the j window, $count_{peak(j)}$ represents the number of peak points of the j window, and $count_{data(j)}$ represents the total number of measurement points of the j window;

3. The peak density standard deviation of all windows is calculated during this period of measurement;

$$s = \sqrt{\frac{\sum_{j=1}^{n}(density(j) - average_{density})^2}{n-1}} \tag{7}$$

4. If there are N segments of complete measurement process, steps 1-3 are performed to get a total of N peak density standard deviations, and the mean is calculated;

5. The alternative integral interval length of 3–30 s is used to perform steps 1–4 respectively, and the maximum integral interval length is the optimal integral interval length.

Using the optimal integral interval length, the measured sequence of $SO_2$ and $CO_2$ is transformed into the average sequence of $SO_2$ and $CO_2$ respectively according to Eq. (2), and all the peak points in the sequence are extracted for subsequent matching.

**3.2 Matching based on DTW**

The concentration of $SO_2$ in the ship plume is generally 0–10 ppm. The concentration of $CO_2$ and $SO_2$ are not in the same order of magnitude of 300–10000 ppm. For these to match using the DTW algorithm, the average sequences of $SO_2$ and $CO_2$ need to be normalized by 0–1 to obtain new $SO_2$ and $CO_2$ sequences (marked Q and C):

$$Q = \left\{\frac{SO_2 data(i) - SO_2 data(min)}{SO_2 data(max) - SO_2 data(min)}\right\} \quad i = 1,2,\dots,n \tag{8}$$

$$C = \left\{\frac{CO_2 data(i) - CO_2 data(min)}{CO_2 data(max) - CO_2 data(min)}\right\} \quad j = 1,2,\dots,n \tag{9}$$

The DTW algorithm calculates the similarity based on the Euclidean distance matrix between two sequences. In the calculation of the Euclidean distance, only one-dimensional numerical value is considered, whereas the selection of the global optimal peak point needs to consider not only the relative size of the concentration, and the time span (the response time deviation of different sensors is usually a few seconds). The matching between two points with a large time span belongs to abnormal matching, so it is necessary to use two-dimensional the relative position relationship to represent the distance between two points. The Manhattan distance considers the values on the two axes, which is more suitable



for the selection of the global optimal peak point than the Euclidean distance. Because the average value
has been normalized, the time span needs to be converted to a value between 0 and 1, so it is replaced by
the ratio of the time span to the total sequence length:
$$d(Q_i, C_j) = \frac{|i-j|}{n} + |y_i - y_j| \qquad (10)$$

In the Manhattan distance calculated following Eq. (10), the proportion of $|y_i - y_j|$ is too large, thus we
add a compensation coefficient to balance the size of the two dimensions:
$$d(Q_i, C_j) = \frac{|i-j|}{n} * 9 + |y_i - y_j| \qquad (11)$$

Through the improved Manhattan distance, a distance matrix of $k * m$ is constructed (the number of $SO_2$
and $CO_2$ peak points is k and m, respectively), which is marked as A, in which the point $(i, j)$ represents
the distance $d(Q_i, C_j)$ between the i point of Q and the j point of C. The smaller the $d(Q_i, C_j)$ value, the
higher the similarity between $Q_i$ and $C_j$. Finding a warping path in the matrix A starts from the (0,0)
point of the matrix to the end of the $(k, m)$ point, and the superposition produces the smallest distance.
The warping path is mainly constrained by the following three aspects: monotonicity, continuity, and
boundary conditions.

1.  The monotonicity constraint means that the warping path can only extend in the prescribed
direction. For a point $(a, b)$ on the path and the next point $(a', b')$ on the path, it must satisfy
$0 \leq (a' - a)$ and $0 \leq (b' - b)$;
2.  The constraint of continuity means that any point in sequence Q and sequence C can be
mapped in a warping path, which can only be aligned adjacent to each other and not matched
across points. For a point $(a, b)$ and the next point $(a', b')$ on the path, it must satisfy
$(a' - a) \leq 1$ and $(b' - b) \leq 1$;
3.  The constraint of the boundary condition means that the first and last point on the warping
path must be $(0,0)$ and $(k, m)$.
Starting from (0,0), each point in the sequence Q and C is matched one by one, and the distance is
accumulated. When the final cumulative distance is reached $(k, m)$, the result is the final distance
measure of the sequence Q and C, that is, the similarity between Q and C. Recursion can be performed
following Eq. (12):
$$\psi(i, j) = d(Q_i, C_j) + min \begin{cases} \psi(i, j-1) \\ \psi(i-1, j-1) \\ \psi(i-1, j) \end{cases} \qquad (12)$$

Each point on the warping path corresponds to a $SO_2$ and $CO_2$ peak points, with a matching relationship
between the two points. However, some of the peak points extracted from the average series are due to


small fluctuations caused by the interference of external environmental factors during the measurement,
which will affect the direction of the warping path when participating in the match. Therefore, the
matching results are not all reasonable, and this is what needs to be selected.
**3.3 Filter matching relationship**
The points of the warping path found by the DTW algorithm in the distance matrix do not have a
reasonable matching relationship. Zhou et al. (2019) previously proposed several criteria to be followed
when selecting the global optimal peak point: one is to eliminate the sharp changes of the peak point,
because these abnormal changes are caused by the uncertainty of the sensor, the monitored gas and its
content in the atmosphere will affect the selection of the global optimal peak point; the other is to
eliminate the peak point with a time span of more than 20 s.
In a completely ideal case, the average sequence of $SO_2$ and $CO_2$ should be in a state of complete
synchronization. When the monitoring equipment slowly approaches the plume of the ship, the average
value of $SO_2$ and $CO_2$ will increase, but when it is far away from the plume, it will be in a decreasing
trend. Because the ideal state will not be disturbed by external environmental factors or the sensor itself,
at any time, the average difference between $SO_2$ and $CO_2$ will remain stable and will not change greatly.
However, in the actual measurement, the complex external environment may make the average difference
at different time points different, hence it cannot be used as the global optimal peak point. Therefore, it
is necessary to find a threshold that can distinguish between normal and abnormal changes, and eliminate
the matching results with a large difference in average. In this study, the K-means clustering algorithm
is used to cluster the difference of the normalized mean of all matching results (marked as D) and obtain
the threshold to distinguish between normal and abnormal changes. The specific process is as follows:
1.   The number of initial cluster centers k is 2, that is, the sample set is divided into two categories,
normal and abnormal changes;

2.   Two data points in D are randomly selected as the initial cluster centers of the two clusters;
3.   The similarity between each sample point and the two cluster centers is calculated, and the
sample points are divided into the clusters corresponding to the cluster centers with the
greatest similarity;

4.   The cluster center of each cluster is recalculated based on the samples in the existing cluster;
5.   Iteration through steps 3 and 4 is performed until the center of the cluster no longer changes.
After multiple k-means clustering of D, if D can be stably divided into two categories, then the threshold
has been found. If the normalized difference of $SO_2$ and $CO_2$ in the matching result is greater than the



threshold, it shows that the matching result belongs to abnormal change and should be eliminated.
According to the above two conditions, the reasonable matching results are screened out, and those
containing the maximum average value of $SO_2$ are retained, while the matching results containing the
maximum average value of $CO_2$ are selected as the global optimal peak point from all the matching
results found.
**3.4 Evaluation of measurement data quality**
Using the above methods, the automatic calculation of emission factors can be realized. However, there
is no suitable method to evaluate the quality of the measured data. Therefore, this study proposes a
method to evaluate the quality of the measured data, according to the index calculation results of the
measured data. In this study, 16 evaluation indexes are proposed, which are sample entropy, information
entropy, third quartile, standard deviation, skewness, standard deviation of peak density, permutation
entropy, fuzzy entropy, approximate entropy, mutual information, first quartile, kurtosis, DTW shortest
distance, quartile spacing, coefficient of variation, and ratio of the number of peak points. Among them,
the shortest distance of DTW, the ratio of the number of peak points, and mutual information are obtained
by calculating $SO_2$ and $CO_2$ measurement data. The rest of the evaluation indexes can be obtained only
by calculating $SO_2$ or $CO_2$. To ensure the accuracy of the evaluation results, for these indexes, we need
to use $SO_2$ and $CO_2$ measurement data, and select the set with higher accuracy to calculate the index. The
role of each evaluation index is shown in Table 1.
**Table 1: Sixteen evaluation indexes and their roles in evaluating data quality.**

| Evaluation index name | Role |
| --- | --- |
| **Sample entropy** | |
| **Information entropy** | Evaluating the random degree of the sequence can be |
| **Permutation entropy** | used to study whether the complexity of the measured |
| **Fuzzy entropy** | sequence is related to the selection of peak points. |
| **Approximate entropy** | |
| **DTW shortest distance** | Measure the relationship between multiple sequences. |
| **Mutual information** | |
| **Ratio of the number of peak points** | Measure the relationship between the difference in the |





| | |
|---|---|
| | number of peak points in a series and the results of global optimal peak point screening. |
| **Standard deviation of peak density** | Reflect the aggregation trend and data change trend of peak points in the series. |
| **First quartile** | |
| **Third quartile** | Reflect the distribution law of measured values. |
| **Quartile spacing** | |
| **Skewness** | Measure the asymmetry of the sequence distribution. |
| **Kurtosis** | Measure the steepness of the sequence distribution. |
| **Standard deviation** | Measure the degree of dispersion of the sequence as a |
| **Coefficient of variation** | whole. |


The uncertainty of the evaluation index can be quantified using the numerical method based on self-
expanding sampling, so as to determine its confidence interval (Wu et al., 2020; Zhong et al., 2007).
Therefore, through 10000 self-developing sampling of the existing measured data and calculating the
average value of each sampling, a set composed of several mean values can be obtained, according to
which the confidence interval of the evaluation index can be obtained. If the index value is in the 95%
confidence interval corresponding to the index, it determines the high-quality of the measured data, which
is marked as 1. If the index value is outside the 95% confidence interval, the measured data is judged as
low-quality data by the index, which is marked as 0. At the same time, when the quality of the measured
data is labeled, the peak trend is obvious, and the high synchronization of the $SO_2$ and $CO_2$ average series
is considered as high-quality data, and the quality label is 1; if the two series have large differences and
drastic changes occur frequently, they are considered as low-quality data, and the quality label is 0.
Combined with the evaluation results of indexes and quality labels, the evaluation accuracy of each index
is calculated, and a certain number of indexes with high accuracy are selected to form an index set to
jointly evaluate the data quality. The distance between the index value of the measured data and overall
mean of the central position of the confidence interval is calculated, and the ratio to the unilateral length
of the confidence interval is obtained. The closer the ratio is to 0, the better the quality of the measured
data. On the contrary, the closer to 1 means the worse the quality of the measured data. When the





numerator in the ratio is greater than the denominator, the ratio is greater than 1, and the index value is
not in the 95% confidence interval. Therefore, the result with a ratio greater than 1 is also reassigned to
1. In joint evaluation, if the calculated ratio of all indexes is 1, the quality of the measured data is judged
to be poor, otherwise the mean value of all ratios less than 1 is closer to 0, indicating a better quality of
the measured data.
**4 Experiment**
**4.1 Data**
Our research team designed and developed a sniffer system using an unmanned aerial vehicle (UAV) in
the "Shanghai Free Trade Zone ship exhaust Monitoring" (MISEE) project. Field tests have demonstrated
that it has the advantages of high monitoring accuracy and convenience (Zhou et al., 2019; Zhou et al.,
2020). In this study, this system is used to collect $SO_2$ and $CO_2$ data of ship emissions. The monitoring
site is located at Waigaoqiao Port in Shanghai, which is only 20 km from the city center. Exhaust gas
measurement data ($SO_2$+$CO_2$) of 148 ships were collected between 2019 and 2021. Six groups of
measurement data are defined to show the rationality and accuracy of the integral interval length selection,
peak point matching and measurement data quality evaluation method. The six groups of data are 2019-
9-27C, 2019-10-17C, 2021-1-13B, 2021-3-10A, 2021-8-18C, and 2021-9-3A. The date in the number is
the date on which the measured data is collected in the field, and the letter indicates the group of data
collected on that date. Among them, the data quality of 2021-1-13B and 2021-3-10A is better, with a
strong synchronization between the two sequences, while the quality of other measured data is poor,
reflecting the peak trend that $SO_2$ and $CO_2$ sequences cannot keep synchronization.
**4.2 Selection of integral interval length**
The result of calculating the optimal integral interval length of 148 groups of measurement data using
the method discussed in 3.1 is shown in Figure 1. The standard deviation of the peak density
corresponding to 12 s is the maximum, that is, 12 s is the optimal integral interval length. Therefore, the
measured value per second is replaced with the average of 12 s, and the average sequence is obtained.






**Figure 1. The 3–30 s integral interval length of the peak density standard deviation of the calculated results.**
**The red point is the result of the calculation corresponding to the length of the 12 s integral interval, taking**
**the maximum value.**

There are differences in the quality of measurement data. High-quality measurement data have a clear
peak trend, low-quality measurement data are mixed with small fluctuations, and the peak trend is not
obvious. In order to verify the effect of using 12 s as the integral interval, the six groups of measurement
data in Figure 2 are taken as an example, which are the measurement data numbered 2019-9-27C, 2019-
10-17C, 2021-1-13B, 2021-3-10A, 2021-8-18C, and 2021-9-3A, the abscissa represents the time point,
and the interval between the two points, or sampling rate, is 1 s. In addition, the data quality is indicated
in the chart title, in which 2021-1-13B and 2021-3-10A are high-quality data, 2019-9-27C, 2019-10-17C,
2021-8-18C, and 2021-9-3A are low-quality data, the left side is the original measurement sequence, and
the right side is the average sequence processed with the 12 s integral interval. The changing trends of
$SO_2$ and $CO_2$ in 2021-1-13B and 2021-3-10A measurement data are basically synchronized. When $SO_2$
reaches the peak point, $CO_2$ will also reach the peak point in the subsequent time. The peak points at 175
formed by small fluctuations, then 210 time points in 2021-1-13B measurement data are integrated into
a larger peak trend after being converted into an average series. In the 2021-3-10A measurement data,
there was a sharp mutation at the 150–165, 190–210, and 300–320 time points of the $SO_2$ sequence,
which changed to a moderate upward and downward trend after processing. At the same time, the
platform values at 130, 165, 210, and 265 time points were also converted into peak points that could
establish a matching relationship. There are many abrupt changes in the $SO_2$ sequence of 2019-9-27C
measurement data, in which the peak trend at the 260 time point becomes more obvious after being
replaced by the average, which can be selected as the global optimal peak. There is no obvious
synchronous change in the $SO_2$ and $CO_2$ average series in 2019-10-17C, 2021-8-18C, and 2021-9-3A





measurement data. The average 2019-10-17C sequence can smooth small fluctuations to some extent,
and it can also transform the steep peak trend formed by drastic changes in 1750–1950 time points into
a relatively gentle trend. The $SO_2$ sequence of 2021-8-18C only has two peak trends at 400, 475 time
points, but because these two peak trends are abrupt, which are converted to average values and flattened
into platform values, and there is almost no synchronization between the two sequences, it is impossible
for this group of low-quality measurement data to select the global optimal peak points. Although 2021-
9-3A has undergone a drastic change, the frequency of the change is not high compared with 2019-10-
17C, hence the effect of the average sequence is more obvious. There are many platform values in the
original $CO_2$ measurement series, such as those in the 0–45 and 52–90 time points. The average sequence
converts all these platform values into non-platform values, which makes the overall change trend clearer.

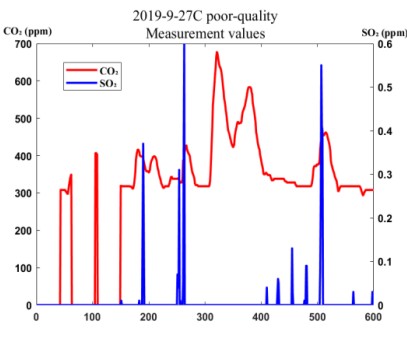
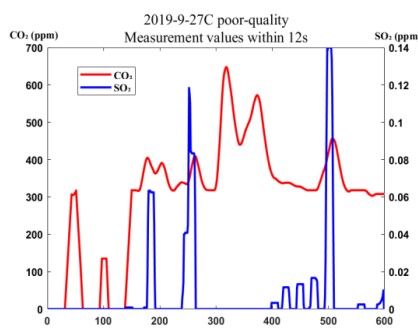

(a)

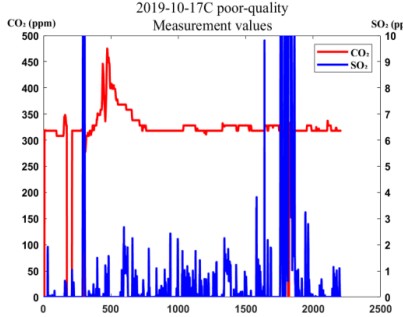
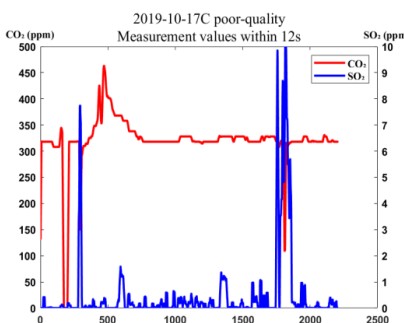

(b)



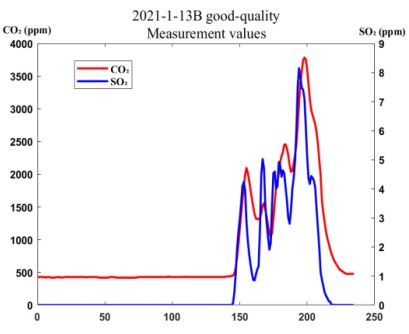
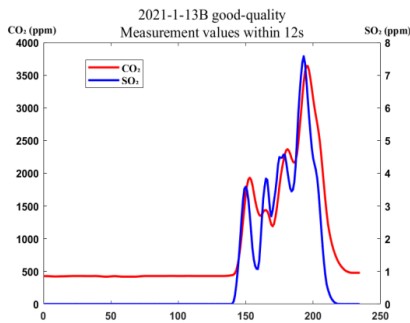

(c)

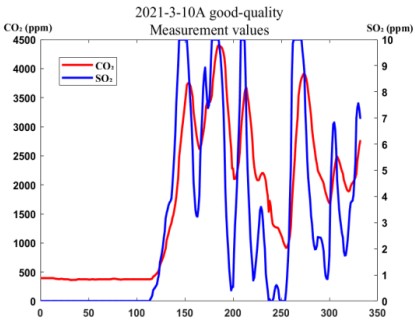
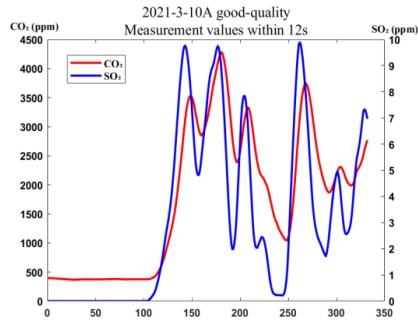

(d)

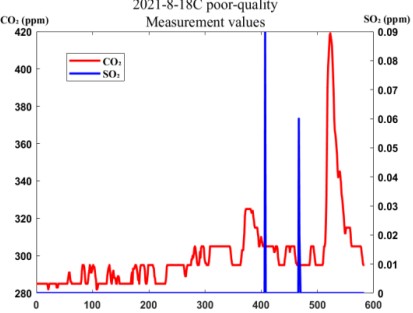
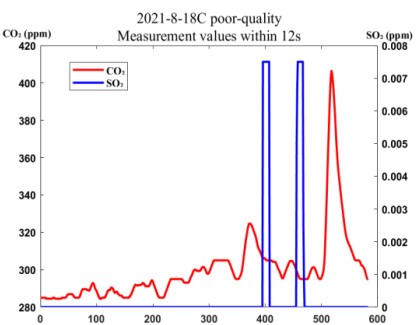

(e)



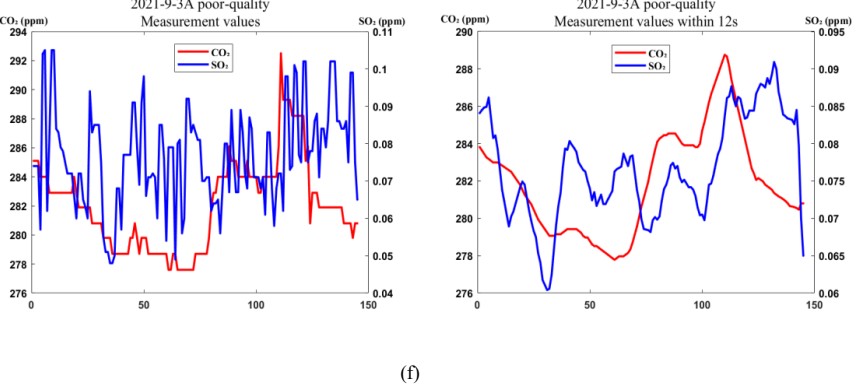

(f)

**Figure 2. Groups of measurement data of $SO_2$ and $CO_2$. (a) 2019-9-27C; (b) 2019-10-17C; (c) 2021-1-13B; (d) 2021-3-10A; (e) 2021-8-18C; and (f) 2021-9-3A. The time and quality of data measurement are marked in the title. On the left is the unprocessed sequence of measurements, and on the right is the average sequence processed with the length of the 12s integral interval.**

**4.3 Matching of peak points and screening of global optimal peak points**

After the measured value series is converted into an average series, and the peak points of $SO_2$ and $CO_2$ are extracted, the corrected time series is transformed into a matching relationship between the peak points of the two average series, and the corresponding peak point of each peak point in another series is found. The DTW algorithm based on the improved Manhattan distance is used to match the peak points of $SO_2$ and $CO_2$.

After the peak point matching is complete, the matching results with a time span of more than 20 s and drastic changes need to be eliminated. To find the threshold to distinguish between normal and abnormal changes, calculations are needed. For this reason, we use 148 sets of measurement data for DTW matching, obtain 911 matching results, calculate the difference between the normalized results of $SO_2$ and $CO_2$ in each matching result, and then use the K-means clustering algorithm for the 911 differences. The clustering result is shown in Figure 3, the result of multiple clustering is stable, and the threshold is 0.3485. Therefore, after the DTW matching for the peak points of a group of measured data, traversing all the matching results, if the normalized difference between $SO_2$ and $CO_2$ is more than 0.3485 or the time span is more than 20 s, cannot be regarded as the global optimal peak point and needs to be eliminated. The matching result containing the maximum average value of $SO_2$ is found among the reserved matching results, and the matching result containing the maximum average value of $CO_2$ is selected as the global optimal peak point from all the matching results found.

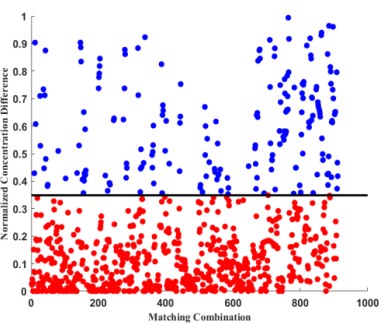

**Figure 3. K-means clustering of 911 normalized differences. The red mark point refers to the normalized difference of the normal change, the blue mark point refers to the normalized difference of the abnormal change, and the black line is the threshold to distinguish normal from abnormal changes.**

The results of peak point matching show that the improved Manhattan distance can consider the important time span and concentration difference. Figure 4 shows the results of three groups of measurement data (2019-9-27C, 2021-1-13B, and 2021-3-10A) screening peak points using Euclidean distance and improved Manhattan distance in the DTW algorithm. The Abscissa represents the time point, and the interval between the two points, or sampling rate, is 1 s. The reasonable global optimal peak point (in the green circle) cannot be found by using the Euclidean distance in 2019-9-27C and 2021-3-10A. The quality of 2019-9-27C is poor, the variation trend of $SO_2$ and $CO_2$ average series is different, and the platform value cannot be used as peak point at the 500 time point. The quality of 2021-3-10A data is good, but the matching relationship between $SO_2$ and $CO_2$ peak point of synchronous change at the 145, 165, and 260 time points are not established, and the normalized concentration of $SO_2$ in the selected global optimal peak point is lower than that of the 260 time point. The reason is that the Euclidean distance only uses the concentration difference to represent the distance of the peak point, and the matching will be given priority when there are other peak points with similar concentration near a peak. The constraint of the warping path leads to the possibility that it can no longer match the later more suitable peak points. The improved Manhattan distance can ensure that the peak points are matched preferentially with similar time span and concentration. The 2019-9-27C and 2021-3-10A groups successfully establish a correct matching relationship between the above peak points when matching using the improved Manhattan distance. As a result, the appropriate global optimal peak point is found. The $SO_2$ and $CO_2$ of the 2021-1-13B measurement data show a trend of almost synchronous change with a peak, hence both the Euclidean distance and the improved Manhattan distance give the appropriate





global optimal peak. However, the Euclidean distance does not establish a matching relationship between
the peak points of this group at the 170 time point, whereas the improved Manhattan distance achieves
this.
Figure 5 shows three groups of measurements with typical problems (2019-10-17C, 2021-8-18C, 2021-
9-3A). There is almost no synchronous change interval between the two time-series of the 2019-10-17C
measurement data, and the peak trend is not clear. When using the screening algorithm, some peak points
with matching relationship are still retained after the time span is more than 20 s and the concentration
difference is more than 0.3485, but these matching results cannot be used as global optimal peak points.
The $SO_2$ measurement data in 2021-8-18C have two platform-like data (data points in the pink circle),
and there is no peak point of $SO_2$, so the global optimal peak point cannot be selected. There are 20 peak
points in the $SO_2$ sequence in 2021-9-3A and only two peak points in the $CO_2$ sequence. Due to the large
difference between the number of peak points, the warping path is easy to shift to the boundary of the
distance matrix, and the global optimal peak point (the peak point in the green circle) cannot establish a
matching relationship. For 2019-10-17C, the screening algorithm gives the global optimal peak point,
but the result is obviously not a suitable global optimal peak point, which is contrary to the subjective
judgment. For 2021-8-18C, the screening algorithm does not recognize a global optimal peak point,
which is consistent with the subjective judgment. For 2021-9-3A, the screening algorithm also recognizes
no global optimal peak point, but it is contrary to the subjective judgment.

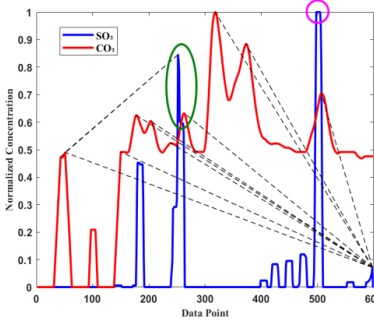
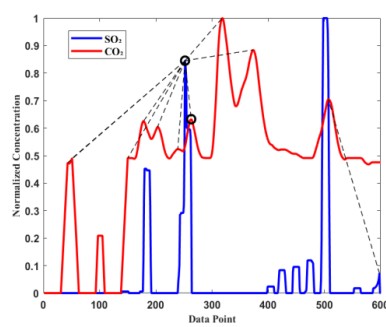

(a)

(b)

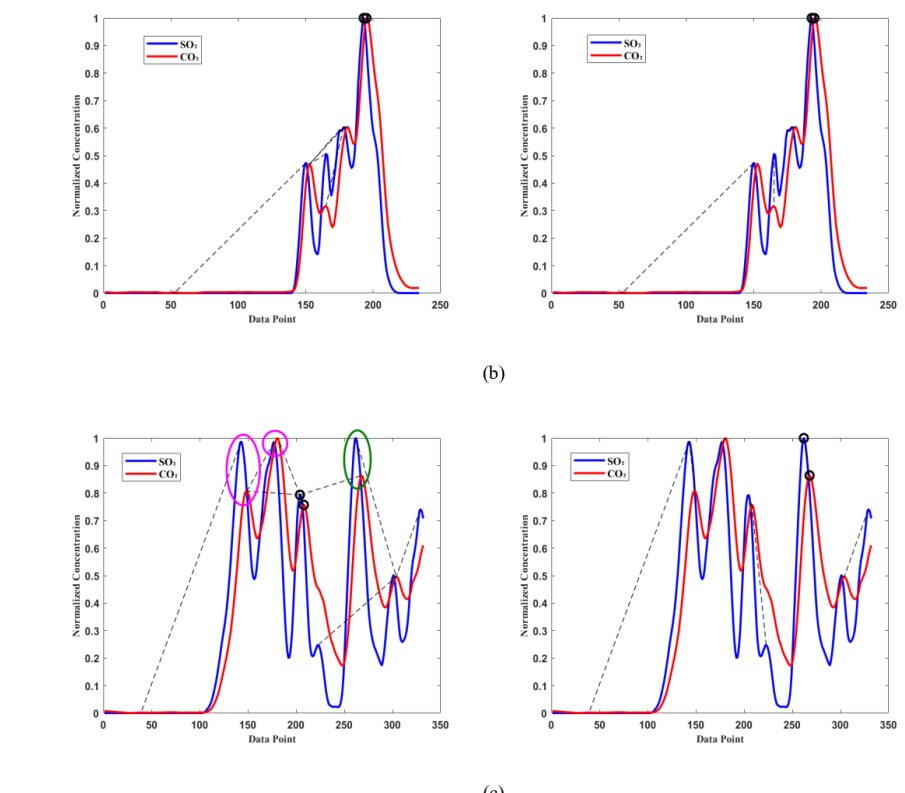

(c)

**Figure 4. Four groups of measurement data of $SO_2$ and $CO_2$. (a) 2019-9-27C; (b) 2021-1-13B; and (c) 2021-3-**
**10A. The left panels are the global optimal peak point screening result using the Euclidean distance, while the**
**right are the global optimal peak point screening using the improved Manhattan distance. The black line is**
**the matching relationship between the two points, the small black circle is the global optimal peak point given**
**by the screening algorithm, the pink circle is the peak points that cannot be regarded as the global optimal**
**peak points, and the green circle is the appropriate global optimal peak points.**

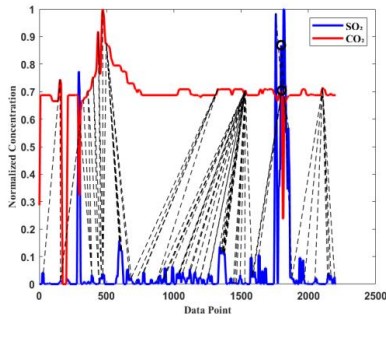

(a)

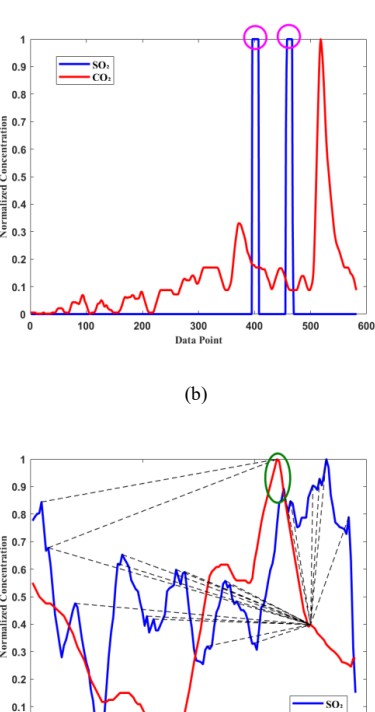

(b)

(c)

**Figure 5. Three groups of SO₂ and CO₂ measurement data. (a) 2019-10-17C, (b) 2021-8-18C, and (c) 2021-9-3A. Abscissa is the point of time, and the interval between the two points, or sampling rate, is 1 s. The black line shows that there is a matching relationship between the two connected peak points, the pink circle is the peak points that cannot be regarded as the global optimal peak points, the green circle is the appropriate global optimal peak points, and the small black circle is the global optimal peak point given by the screening algorithm.**

Of the 148 groups of measurement data screened, 101 groups, in which the artificial screening results are consistent with the algorithm screening, demonstrate the global optimal peak points. Among the remaining data, 19 groups of measurement data show inconsistent results, while 28 groups have poor quality that neither screening could find the global optimal peak points. Regardless of whether the global optimal peak point exists or not, the result of the algorithm is defined as correct when the results of the subjective and algorithm screenings are the same, in this case the correct rate of the screening algorithm is 87.16%. The results of the screening algorithm are consistent with the subjective screening results, but the global optimal peak points are not found, or the large deviation of time series leads to no reasonable





retention of peaks in the screening matching relationship. For this measurement data, the algorithm can
identify and provide the reason. There are mainly two cases of inconsistent measurement data, one is that
the data can artificially screen the global optimal peak, yet the algorithm does not give the optimal result
or directly determines that there is no suitable global optimal peak, the other is that there is no global
optimal peak point in the data, and the result given by the algorithm is only a group of peak points where
the concentration difference and time span do not exceed the threshold. The first case usually occurs in
the measurement data with a large gap in the number of peak points in the average sequence of $SO_2$ and
$CO_2$. When forming a distance matrix, this difference will cause one side of the row or column of the
matrix to be much larger than the other. Limited by the directional constraints of the DTW algorithm,
there is no other choice for the warping path to reach the matrix boundary, but can only extend vertically
or horizontally along the boundary. Therefore, the appropriate global optimal peak point will not be
selected by the warping path. The second situation is more common. Multiple screening conditions are
defined in the screening matching relationship, and the matching relations that do not meet the conditions
will be eliminated. Although the remaining matching relations meet the conditions, they may not
necessarily be the optimal peak points. This situation will occur when the algorithm selects the final
value from the reserved peak value.
**Table 2: Algorithm screening for groups of measurement data. "Consistent (found)" means that the result**
**given by the algorithm is the same as that of artificial selection and can find the global optimal peak point.**
**"Consistent (not found)" means that the two results are consistent and neither of them has found the global**
**optimal peak point. "Inconsistent" refers to the difference between the two results. "Correct rate" refers to**
**the ratio of the 148 groups of measurement data to the 148 groups of data with "Consistent (found)" and**
**"Consistent (not found)" measurement data.**

| Consistent (found) | Consistent (not found) | Inconsistent | Correct rate |
|---|---|---|---|
| 129 | 28 | 19 | 87.16% |

**4.4 Verification of data quality evaluation method**
In Section 3.4, 16 indexes are proposed to evaluate the quality of the measured data. The 95% confidence
interval of the evaluation index is calculated by 10000 self-development sampling of the collected data,
and the results are shown in Table 3.
**Table 3: Self-development sampling times of 16 evaluation indexes and the upper and lower bounds of the 95%**
**confidence interval.**



| Evaluation index | Self-development sampling times | Lower bound of 95% confidence interval | Upper bound of 95% confidence interval |
|---|---|---|---|
| Sample entropy ($SO_2$) | 10000 | 0.082153984 | 0.150936086 |
| Information entropy ($SO_2$) | 10000 | 3.245317702 | 3.721704023 |
| Third quartile ($SO_2$) | 10000 | 0.203210727 | 0.280760482 |
| Standard deviation ($SO_2$) | 10000 | 0.211572244 | 0.22958644 |
| Skewness ($CO_2$) | 10000 | 1.027746851 | 1.471870163 |
| Standard deviation of peak density ($CO_2$) | 10000 | 0.040255915 | 0.045275922 |
| Permutation entropy ($SO_2$) | 10000 | 0.694783281 | 0.752033377 |
| Fuzzy entropy ($CO_2$) | 10000 | 0.179056426 | 0.216571182 |
| Approximate entropy ($CO_2$) | 10000 | 0.140398491 | 0.167099422 |
| Mutual information | 10000 | 0.588883766 | 0.647985481 |
| First quartile ($SO_2$) | 10000 | 0.042864083 | 0.089750709 |
| Kurtosis ($CO_2$) | 10000 | 5.435791486 | 7.228334146 |
| DTW shortest distance | 10000 | 14.44706187 | 26.07445393 |
| Quartile spacing ($SO_2$) | 10000 | 0.152163801 | 0.207432999 |
| Coefficient of variation ($SO_2$) | 10000 | 1.651178674 | 1.980617162 |
| Ratio of the number of peak points | 10000 | 1.996089804 | 6.083568681 |

The results of the 16 evaluation indexes for 148 groups of measurement data are shown in Table 4. Some
indexes are selected to form the index set. Skewness ($CO_2$) and information entropy ($SO_2$) with the
highest accuracy of index evaluation are taken as the elements in the initial subset. After comparing the
accuracy of adding other indexes, it is found that when the elements in the index set are skewness ($CO_2$),
information entropy ($SO_2$), sample entropy ($SO_2$), and quartile spacing ($SO_2$), the joint evaluation can
achieve relatively good results, with a correct rate of up to 70.95%. When the index continues to increase,
the correct rate decreases, indicating that these indexes can only represent part of the characteristics of
the measured data and have high coupling. Therefore, these four indexes are selected for joint evaluation.
The six groups of measurements in Figure 6 have all been defined. This section uses the quality
evaluation method to evaluate the quality and verify the rationality of the six groups of data. The groups





2019-9-27C, 2021-1-13B, and 2021-3-10A are the measurements that can screen out the global optimal
peak points, but the $SO_2$ and $CO_2$ series in 2019-9-27C do not have a synchronous trend, and only the
two series have the same peak trend at 2019-9-27C, thus even if they can be used as global optimal peak
points, the ratio of the four evaluation indexes is 1, and the joint evaluation result is poor quality. There
are four peak trends in 2021-1-13B, where $SO_2$ and $CO_2$ change almost synchronously. The peak trend
at the 200 time point is very clear and higher than other peaks, which belongs to the appropriate global
optimal peak point, and the result of joint evaluation of indexes is very close to 0. The peak trend in
2021-3-10A is slightly more than that of 2021-1-13B, although altogether is basically synchronous; the
peak trend at 145, 160, 260 time points can be observed, while between the global optimal peak point is
the priority to select the maximum value of the average $SO_2$. Finally, the peak point at the 260 time point
is selected as the global optimal peak point, and the joint evaluation result of the index is higher than that
of 2021-1-13B. The $SO_2$ average sequence of 2019-10-17C has frequent small fluctuations and abrupt
changes over time, and there is no obvious synchronous peak trend, while the $SO_2$ sequence of 2021-8-
18C has platform values only at 400, 480 time points, thus it cannot be used as the global optimal peak
point. The average series of $SO_2$ and $CO_2$ in 2021-9-3A show a synchronous change trend at 90–110 time
points, hence $SO_2$ fluctuates greatly relative to $CO_2$. Although the peak point at the 110 time point can
be selected as the global optimal peak, and the result of joint evaluation of the index remains 1. To
summarize, the results of the quality evaluation methods for poor and good quality data are in line with
expectations.
**Table 4: Sixteen indexes for the evaluation of 148 groups of measurement data statistics.**

| Index | Number of correct evaluation results | Number of wrong evaluation results | Correct rate |
|---|---|---|---|
| **Sample entropy ($SO_2$)** | 86 | 62 | 58.11% |
| **Information entropy ($SO_2$)** | 90 | 58 | 60.81% |
| **Third quartile ($SO_2$)** | 87 | 61 | 58.78% |
| **Standard deviation ($SO_2$)** | 86 | 62 | 58.11% |
| **Skewness ($CO_2$)** | 91 | 57 | 61.49% |
| **Standard deviation of peak density ($CO_2$)** | 82 | 66 | 55.41% |



| | | | |
|---|---|---|---|
| **Permutation entropy (SO₂)** | 81 | 67 | 54.73% |
| **Fuzzy entropy (CO₂)** | 82 | 66 | 55.41% |
| **Approximate entropy (CO₂)** | 87 | 61 | 58.78% |
| **Mutual information** | 76 | 72 | 51.35% |
| **First quartile (SO₂)** | 82 | 66 | 55.41% |
| **Kurtosis (CO₂)** | 85 | 63 | 57.43% |
| **DTW shortest distance** | 73 | 75 | 49.32% |
| **Quartile spacing (SO₂)** | 88 | 60 | 59.46% |
| **Coefficient of variation (SO₂)** | 87 | 61 | 58.78% |
| **Ratio of the number of peak points** | 74 | 74 | 50.00% |

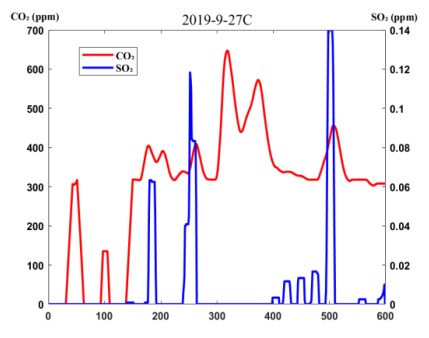

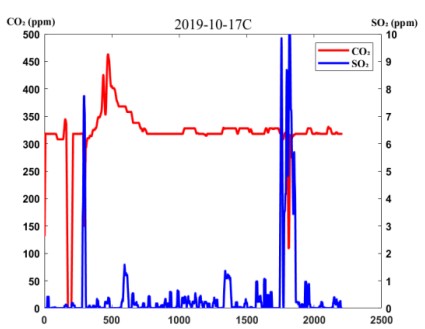

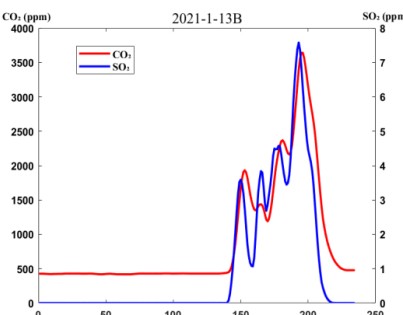

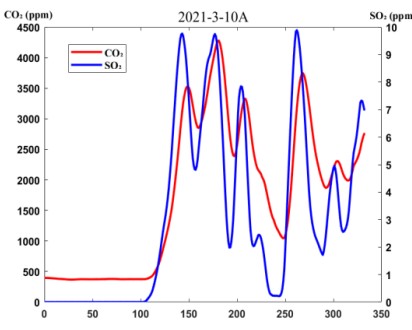



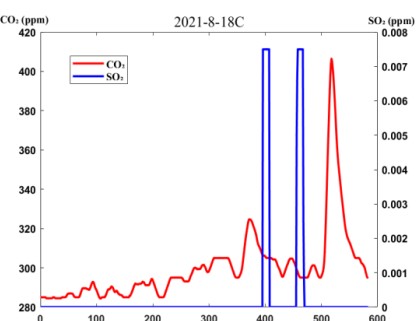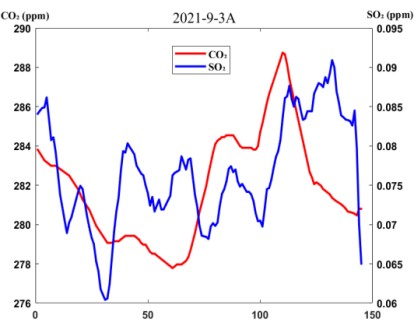

**Figure 6. Evaluation of the quality of the six groups of $SO_2$ and $CO_2$ measurement data. The Abscissa is the time point, and the interval between the two points, or sampling rate, is 1s.**

**Table 5: Algorithm screening of six groups of measurement data (2019-9-27C, 2019-10-17C, 2021-1-13B, 2021-3-10A, 2021-8-18C, and 2021-9-3A). The specific values and evaluation results of information entropy ($SO_2$), sample entropy ($SO_2$), skewness ($CO_2$), quartile spacing ($SO_2$) and the results of joint evaluation of the four indexes.**

| Data ID | Algorithm | Information entropy | | Sample entropy | | Skewness | | Quartile spacing | | Result of joint evaluation |
|---------|-----------|-------|------|-------|------|---------|------|--------|------|------------|
| | | Value | EVAL | Value | EVAL | Value | EVAL | Value | EVAL | |
| **2019-9-27C** | √ | 1.7112 | 1 | 0.0307 | 1 | -0.6145 | 1 | 0.0060 | 1 | 1 |
| **2019-10-17C** | × | 2.6154 | 1 | 0.0353 | 1 | -3.8664 | 1 | 0.0240 | 1 | 1 |
| **2021-1-13B** | √ | 2.6865 | 1 | 0.0272 | 1 | 1.6976 | 1 | 0.1801 | 0.0116 | 0.0116 |
| **2021-3-10A** | √ | 5.1502 | 1 | 0.1061 | 0.3050 | 0.2466 | 1 | 0.5737 | 1 | 0.3050 |
| **2021-8-18C** | √ | 0.3047 | 1 | 0.0076 | 1 | 3.1325 | 1 | 0 | 1 | 1 |
| **2021-9-3A** | × | 5.8638 | 1 | 0.7894 | 1 | 0.4692 | 1 | 0.3711 | 1 | 1 |

**"×" indicates that the screening algorithm is inconsistent with subjective judgment**

**5 Conclusion**

The emission factor is an important parameter for compiling ship emission inventory. Various studies

have measured emission factors under different conditions. However, there is little attention and research

on improving the accuracy, confidence, and automation of the calculation methods.

In this study, we propose a high accuracy calculation of ship emission factors and an evaluation of the





quality of measurement data based on the sniffer method. Altogether, we optimize the calculation process,
select the appropriate integral interval length to convert the original measurement data into an average
value to make the calculation results more stable, and put forward the concept of peak density standard
deviation selection for the optimal integral interval length, which reduces the uncertainty caused by the
use of empirical values. The average sequence is extracted to find all the peak points, and the matching
relationship with the measured series of $SO_2$ and $CO_2$ is found by using the DTW algorithm based on the
improved Manhattan distance. The unreasonable matching results are eliminated according to the
concentration change and time span thresholds analyzed. The maximum concentration of $SO_2$ and $CO_2$
in the remaining matching relationship is the global optimal peak point. In order to evaluate the quality
of the measured data objectively, 16 evaluation indexes, which can reflect the characteristics of the
measured data are selected, and the 95% confidence interval of each index is calculated by self-expanding
sampling of the measured data, and combined with the quality label of the measured data. Several indexes
with high accuracy of data quality evaluation are selected to jointly evaluate the measured data, further
optimizing the accuracy of the evaluation.
We used 148 groups of sniffer measurements collected between 2019 and 2021 to test the rationality of
the above methods. When the length of the integral interval is 12 s, the standard deviation of peak density
reaches the maximum. Compared with the change trend of several groups of data before and after
preprocessing, the peak trend of the data set after preprocessing is more obvious, and most of the
nonsignificant trends in the series are smoothed out. Comparing the artificial and algorithm screening
results of the global optimal peak points of 148 groups of measured data, 129 groups have the same
results, with a correct rate of 87.16%. After 10000 times of self-development sampling, the 16 evaluation
indexes all reached the 95% confidence interval. When using four evaluation indexes: information
entropy ($SO_2$), sample entropy ($SO_2$), quartile spacing ($SO_2$), and skewness ($CO_2$), 70.95% of the data
quality labels are consistent with the joint evaluation results of the evaluation index set. When using these
four indexes to quantify data quality, it is consistent with expectations, which verifies that the data quality
evaluation method is feasible.
The emission factor calculation and measurement data quality evaluation methods proposed in this study
can reduce the uncertainty in the current sniffer technique monitoring ship emission research, objectively
evaluate the measurement data quality, and provide data support for the accurate establishment of an
emission inventory. Future work is needed to further analyze the characteristics of the measured data so



as to improve the global optimal peak point screening algorithm. At the same time, it is also necessary
to find other evaluation indexes that can reflect the characteristics of the measured data to reduce the
coupling in the evaluation index set, improve the accuracy of the joint evaluation index set and
quantification of the quality of the data.
**Data availability**
Please address requests for data sets and materials to Fan Zhou (fanzhou_cv@163.com).
**Author contribution**
LZ analyzed the experimental data and authored the article. FZ designed the study and provided
constructive comments on this research.
**Competing interests**
The authors declare that they have no conflict of interest.
**Acknowledgements**
This research was supported by the National Natural Science Foundation of China (grant No. 41701523)
and Science and Technology Commission of Shanghai Municipality (grant No. 22692107400), as well
as the Shanghai High-level Local University Innovation Team (Maritime Safety & Technical Support).

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
