# Peer review of "High accuracy calculation and data quality evaluation of ship emissions based on the sniffer method"

_Atmospheric Chemistry and Physics, 2022_

## Author Comment (AC2)

Answer to Referee #1

We would like to thank Referee #1 for his/her positive and constructive comments and suggestions. We have studied comments carefully and made corrections, which we hope meet with approval. Comments and responses are listed as follows. In order to facilitate the reference to the questions and proposed changes, we use the following color coding:

Color coding:

**Referee comment**

Our answer

Proposed change in manuscript
* * *
**1.Shipping emissions are a relevant source of pollution in particular in the clean marine boundary layer. A better characterisation of ship emissions is a relevant topic for atmospheric chemistry studies and for the control of environmental regulations, and improving the accuracy of emission estimates is necessary for a better understanding of the relevance of ship emissions.**

**In this manuscript, the authors propose a complex algorithm to match air-borne in-situ observations of CO2 and SO2 in ship emission plumes with the aim to determine SO2 emission ratios. Briefly, the algorithm consists of 1) the smoothing of the time series, 2) the assignment of matching peaks in the CO2 and SO2 time series 3) the identification and removal of invalid matches and 4) a quality estimate for the remaining matches. The algorithm is described and applied to a small sample of real measurements of variable quality, and good performance of the algorithm is found in comparison to a subjective assignment.**

**Unfortunately, I cannot recommend this manuscript for publication. The reason for this assessment is that for each of the steps performed, I disagree with the assumptions made and the approaches taken. In many places, ad hoc parameters are introduced and conclusions are drawn without good justification. In addition, the description is hard to follow and often unclear, both because of issues with the use of English and the way the text is written.**

**If I understood the manuscript well, the authors try to identify the matching maxima in the time series of SO2 and CO2 for an individual plume measurement in order to determine the SO2 / CO2 ratio from this value. However, I do not see the rationale for using this specific value. The ratio should be constant throughout the plume and therefore could be computed from any pair of matching measurements. If this is not possible for signal to noise reasons, the ratio of the integrals of the two quantities over the plume transect should be taken as this will be less sensitive to noise and to small time lags between the measurements. I therefore think that the whole idea of finding the maxima in the time series is unnecessary and actually not asking the right question.**

First of all, thank you very much for your interest in our work. Your understanding of the research content designed in this manuscript is very accurate, which makes your views very targeted and valuable. I think the main problem is that this manuscript may have been studied for the wrong purposes, that is, the emission factor can be calculated by "the ratio should be constant throughout the plume and therefore could be computed from any pair of matching measurements", rather than requiring a complex algorithm to handle the problem. In this regard, I would like to make the following explanation:

I think we can all agree that accurate emission factors can be calculated from measurements of stable air flows. However, in

the actual measurement activities, due to the interference of various environmental factors, the data acquired by the "sniffer" equipment is often not the measured value of stable airflow, or the measured value of stable airflow and unstable airflow are mixed together. For this reason, a quantitative and objective method is needed to identify the measurement of stable airflow.

In response to your positive suggestions, we have done the following:

**1. Necessary modifications were made to some discussions in the manuscript to express the methodological process more clearly.**

**2. Published all our data and codes for common discussion and demonstration. The data and code used in this research are available from the Zenodo data repository:** https://www.zenodo.org/record/7831708#.ZD_Xd3ZBxD8

**3. Based on the manuscript, an appendix is added to demonstrate the four steps of this research method. In addition, we also compare the results of other methods.**

We have added the following in the appendix to illustrate the purpose of the research.

From 2018 to 2019, we carried out a number of experiments regarding the monitoring of ship exhaust gas using unmanned aerial vehicles (UAVs) and cooperated with maritime authorities to board ships and extract fuel samples for comparison and verification. Thus, our data include both exhaust measurements of the target ship and measurements of the fuel sulfur content (FSC), which is considered to be the actual value. After combustion, the main source of sulfur in ship fuel is $SO_2$. Therefore, the true FSC value can be converted into the true $SO_2$ emission factor (EF) value on the premise of ignoring the conversion of sulfur into other compounds. As a result, we have both the actual value of the $SO_2$ EF and gas measurements of "$SO_2$+$CO_2$". By analyzing these data, we found that using the gas measurements of "$SO_2$+$CO_2$" in different time series to calculate the $SO_2$ EF and FSC would yield different results, with large differences compared to the actual values. However, in previous research we used empirical methods to select and calculate these values and obtained relatively accurate results. Nevertheless, the importance of the EF accuracy is self-evident, and an objective, accurate and automatic calculation method is urgently needed.

**2.For the smoothing, the authors try to establish an objective criterion by maximising the "peak density standard deviation".**

**However, while this criterion is quantitative, it is not clear to me why this value should lead to the optimum smoothing.**

We have added the following in the appendix to illustrate the effect of this step.

The actual data monitored by two groups of UAVs (plume number 2019-4-12B and 2019-4-15A) are taken as examples to illustrate this problem. The measured value sequence of the two data groups is shown in Figure S1. The change trend of the measured value sequences of $SO_2$ and $CO_2$ in 2019-4-12B are similar, and the obvious peak trend is at time point 620. Therefore, the data quality of this group is high. However, the variation trend in the measured values of $SO_2$ and $CO_2$ in 2019-4-15A is quite different. A steep peak trend appears multiple times in $SO_2$, such as at time points 60, 95, and 190, and the peak trends of $SO_2$ and $CO_2$ are unclear except for a close peak trend at time point 280. This may be due to interference with the actual measurement process caused by the surrounding ship exhaust gas, wind speed, and the direction in the natural environment; thus, the quality of this group of measurement data is low.

In Table S1 and S2, the measured values at 30 time points in the peak areas of 2019-4-12B and 2019-4-15A are listed. Different time spans are selected to calculate the FSC, and the results are compared with the actual FSC values measured on board the ship. This experiment verifies the importance of choosing an appropriate time span according to the difference between the calculated results and the actual values.

As seen in Table S1, the calculation results of three different time spans (10s, 20s, and 30s) are 0.069% (m/m), 0.050% (m/m), and 0.050% (m/m), respectively. When the true value is 0.080% (m/m), the deviation of the calculation result of the 10s time span is -0.011% (m/m), and the deviation of the FSC results corresponding to the 20s and 30s time spans is -0.030%(m/m), which is greater than the former. As for Table S2, the calculation results of the three different time spans (10s, 20s, and 30s) are quite different, namely 0.019% (m/m), 0.016% (m/m), and 0.012% (m/m), respectively. When the true value is 0.044% (m/m), the maximum deviation is -0.032% (m/m) and the corresponding time span is 30s.

However, the upper FSC limit of the ECA is 0.1%(m/m) and this degree of deviation should not be ignored. More importantly, since both the FSC and EF are calculated based on the ratio of $SO_2$ to $CO_2$, it is highly likely that there will be a larger deviation at a higher concentration. To sum up, it is necessary to put forward an objective and reasonable standard and process for a variety of gas measurement series.

[Figure]

**Figure S1. $SO_2$ and $CO_2$ time series of measured data of the 2019-4-12B and 2019-4-15A groups. The actual FSC values are 0.08%(m/m) and 0.044%(m/m), respectively.**

**Table S1. Measured values of $SO_2$ and $CO_2$ of 2019-4-12B at 30 continuous time points. Time spans of 10s, 20s, and 30s are selected to calculate the FSC, and the deviation from the true FSC value (0.08%(m/m)) is listed.**

| Time | $SO_2$/ppm | $CO_2$/ppm | Time span (length) | Estimated FSC/%(m/m) | Deviation/%(m/m) |
|---|---|---|---|---|---|
| 10:54:26 | 1.04 | 3778 | | | |
| 10:54:27 | 1.05 | 3840 | | | |
| 10:54:28 | 1.01 | 3898 | | | |
| 10:54:29 | 0.89 | 3944 | | | |
| 10:54:30 | 0.77 | 3963 | 10:54:36-10:54:45 | 0.069 | -0.011 |
| 10:54:31 | 0.72 | 3965 | (10s) | | |
| 10:54:32 | 0.72 | 3924 | | | |
| 10:54:33 | 0.67 | 3882 | | | |
| 10:54:34 | 0.62 | 3833 | | | |
| 10:54:35 | 0.62 | 3772 | | | |
| 10:54:36 | 0.72 | 3713 | | | |
| 10:54:37 | 0.89 | 3664 | | | |
| 10:54:38 | 1.26 | 3632 | | | |
| 10:54:39 | 1.39 | 3656 | | | |
| 10:54:40 | 1.41 | 3693 | 10:54:32-10:54:51 | 0.050 | -0.030 |
| 10:54:41 | 1.36 | 3764 | (20s) | | |
| 10:54:42 | 1.26 | 3840 | | | |
| 10:54:43 | 1.2 | 3921 | | | |
| 10:54:44 | 1.02 | 3996 | | | |
| 10:54:45 | 0.84 | 4100 | | | |
| 10:54:46 | 0.67 | 4109 | 10:54:26-10:54:55 | 0.050 | -0.030 |
| 10:54:47 | 0.45 | 4069 | (30s) | | |
| 10:54:48 | 0.34 | 3988 | | | |

| 10:54:49 | 0.31 | 3915 |
|---|---|---|
| 10:54:50 | 0.34 | 3806 |
| 10:54:51 | 0.53 | 3596 |
| 10:54:52 | 0.62 | 3499 |
| 10:54:53 | 0.6 | 3415 |
| 10:54:54 | 0.55 | 3340 |
| 10:54:55 | 0.47 | 3265 |

**Table S2. Measured values of $SO_2$ and $CO_2$ of 2019-4-15A at 30 continuous time points. Time spans of 10s, 20s, and 30s are selected to calculate the FSC, and the deviation from the true FSC value (0.044%(m/m)) is listed.**

| Time | $SO_2$/ppm | $CO_2$/ppm | Time span (length) | Estimated FSC/%(m/m) | Deviation/%(m/m)) |
|---|---|---|---|---|---|
| 9:18:15 | 0 | 552 | | | |
| 9:18:17 | 0 | 526 | | | |
| 9:18:18 | 0 | 521 | | | |
| 9:18:18 | 0 | 533 | | | |
| 9:18:19 | 0 | 564 | 9:18:24-9:18:34 | 0.019 | -0.025 |
| 9:18:20 | 0 | 610 | 10s | | |
| 9:18:21 | 0.05 | 671 | | | |
| 9:18:22 | 0.12 | 739 | | | |
| 9:18:23 | 0.14 | 876 | | | |
| 9:18:25 | 0.26 | 1021 | | | |
| 9:18:24 | 0.2 | 933 | | | |
| 9:18:26 | 0.3 | 1124 | | | |
| 9:18:27 | 0.29 | 1241 | | | |
| 9:18:28 | 0.24 | 1356 | | | |
| 9:18:29 | 0.11 | 1540 | 9:18:20-9:18:39 | 0.016 | -0.028 |
| 9:18:30 | 0.04 | 1590 | 20s | | |
| 9:18:31 | 0 | 1614 | | | |
| 9:18:32 | 0 | 1610 | | | |
| 9:18:33 | 0 | 1589 | | | |
| 9:18:34 | 0 | 1565 | | | |
| 9:18:36 | 0 | 1470 | | | |
| 9:18:36 | 0 | 1411 | | | |
| 9:18:37 | 0 | 1349 | | | |
| 9:18:38 | 0 | 1285 | | | |
| 9:18:39 | 0 | 1224 | 9:18:15-9:18:44 | 0.012 | -0.032 |
| 9:18:40 | 0 | 1165 | 30s | | |
| 9:18:41 | 0 | 1110 | | | |
| 9:18:42 | 0 | 1014 | | | |
| 9:18:43 | 0 | 962 | | | |
| 9:18:44 | 0 | 911 | | | |

Based on the results presented in Table S1 and Table S2, we verified the influence of the selection of the integral interval length on the calculation results of the FSC by analyzing the actual measured data. Subsequently, the concept of the "standard deviation of peak density" is proposed in this study to screen for the appropriate length of the integral interval. In the measurement data of the time series, the trend of the peak value is the most important factor. The length of the integral interval with the standard deviation of the maximum peak density indicates that the change in peak density is the most drastic after the length is converted into the mean series, and the trend in the original measurement series is the clearest, avoiding error smoothing due to the conversion process.

**3.The next step is the matching of the peaks in the two time series. The authors use a "Dynamic time warping algorithm" with the Manhatten distance in a normalised time-concentration plane to find matching pairs. However, it is not at all clear why a) this distance is a good metric for finding the peaks and b) why taking the absolute values in the Manhatten distance makes more sense than computing the Eularian distance. More importantly, the need for the matching arises from differences in the time response of the SO2 and CO2 measurements, which should be constant over the short time periods needed to measure a plume. Therefore, the complex algorithm which assigns peaks in the time series allowing variable time lags results in unphysical assignments. It would have been much simpler and a better representation of the physical effects behind the shift in the time series to calculate cross-correlations of the two time series for different realistic time lags to determine the optimal time shift to be applied to the time series.**

We have added the following in the appendix to illustrate the effect of this step.

The Dynamic Time Warping (DTW) algorithm was used to normalize the sequence of the $SO_2$ and $CO_2$ measurements to avoid the effect of the difference in the orders of magnitude of $SO_2$ and $CO_2$. However, the Euclidean distance is the length of a straight line connecting a point on the $SO_2$ sequence to a point on the $CO_2$ sequence. The Manhattan distance is the distance between two points calculated using the difference between two coordinate dimensions. Comparatively speaking, the Manhattan distance can better reflect the actual distance in the time and body measurement dimensions. In addition, it is more convenient to adjust the influence coefficients of two dimensions when the Manhattan distance is used, which is more conducive to the matching of feature points in the DTW algorithm. The calculation results of 2021-5-21A and 2019-7-12A illustrate that the Manhattan distance is more suitable.

2021-5-21A: The overall peak trend is very clear, the two sequences of $SO_2$ and $CO_2$ reach the peak point at approximately the same time, which is less affected by external interference factors, and the data quality is good. The matching results using the Euclidean and Manhattan distances are shown in Figure S2 (a) and (b). Since the number of overall peaks is not high, the matching results are clear. In Figure S2 (a), the combination of peak points at time point 210 is much higher than that at time point 440 and should be used as the global optimal peak point. However, the Euclidian distance does not establish a matching relationship between this peak point pair, while the Manhattan distance in Figure S2 (b) selects the correct result. The reason for this is that the DTW algorithm itself uses three monotone, continuity, and boundary constraints to restrict the direction of the structured path. When the Euclidean distance is used to create a distance matrix, the distance between the peaks of the two sequences cannot be accurately represented, resulting in the structured path deviating from the ideal direction of travel, so that the matching result of the optimal peak is affected by that of other peaks. Therefore, the global optimal peak cannot be accurately selected.

2019-7-12A: Overall, there are many peaks and the data quality is worse than that of 2021-5-21A. However, the peak point combination at time point 410 is evidently superior to other peak trends in terms of time and concentration distance and should be regarded as the global optimal peak point. However, in the Euclidean distance matching results shown in Figure S2 (c), most of the time distance differences are too large. The Manhattan distance shown in Figure S2 (d) solves this problem and its matching results are the combination of peak points with similar time and concentration distances.

Specific constraint conditions:

1. The constraint of monotonicity makes the structured path extend in the specified direction. For a point on the path $(a, b)$, there is a following point on the path $(a', b')$, and $0 \leq (a' - a)$ and $0 \leq (b' - b)$.

2. The constraint of continuity means that any point in two sequences can be mapped to the regular path. They can only be aligned adjacent to each other and cannot be matched across points. For a point on the path $(a, b)$, there is a following point on the path $(a', b')$, and $(a' - a) \leq 1$ and $(b' - b) \leq 1$.

3. Suppose the distance matrix has $k$ rows and $m$ columns, then the boundary property constraint makes the first and last points on the structured path $(0,0)$ and $(k, m)$.

By comparing the matching results of these two data sets, it can be seen that the Euclidean distance cannot reflect the importance of the time and concentration distances when measuring the distance between measured values. Therefore, it is

impossible to construct the distance matrix directly from the distance between two points. The Manhattan distance considers the time and the concentration distance in an appropriate way, so as to reduce the frequency and probability of abnormal matching results as much as possible. The matched results are the combination of peak points with small time and concentration distance gaps. Therefore, the Manhattan distance is selected for the matching process in this study.

[Figure]

(a) DTW matching based on Euclidean distance      (b) DTW matching based on Manhattan distance

(c) DTW matching based on Euclidean distance      (d) DTW matching based on Manhattan distance

**Figure S2. Two sets of measured data of the 2021-5-21A and 2019-7-12A groups and their DTW matching results using the Euclidian and Manhattan distance. The measurement value highlighted by the black circle is the global optimal peak identified by the algorithm and the black dashed line is a pair of peak points matched by the algorithm, including an $SO_2$ and a $CO_2$ measurement value.**

**4.As the algorithm proposed in the manuscript is not well constrained (see the last point), it often produces unphysical results. In order to identify and remove them, the authors propose a k-means clustering of the "Normalised Concentration Differences" with two clusters. Why such an approach should be able to separate valid and invalid results is not clear. In my opinion, a simple threshold removing pairs with "too large" differences would lead to similar results and be equally subjective.**

We have added the following in the appendix to illustrate the effect of this step.

In the process of using UAVs equipped with "sniffer" monitoring equipment to collect ship exhaust samples and obtain measurement values of $SO_2$ and $CO_2$, a variety of uncertain factors will cause interference, which may come from the external environment or from the monitoring process. The former mainly includes wind direction, wind speed, air temperature, and atmospheric $SO_2$ and $CO_2$ stocks, among others, while the latter can be summarized as the uncertainties introduced by the gas sensor, measurement, calculation, and exhaust. Due to these uncertainties, the data quality of the $SO_2$ and $CO_2$ measurement series will be reduced by a certain extent, which is mainly reflected in the unclear peak trend. When the DTW algorithm is used to match the peak points, the direction of the structured path is restricted by the three boundary, continuity, and monotonicity constraints, which enables it to establish a matching relationship among all peak points to be matched without missing any. However, it is precisely because of the existence of such constraints that unsatisfactory matching results will appear in the case of poor-quality data. One of the most

common cases is that after the structured path reaches the right- or bottommost boundary, the remaining peak points can only be matched along a single direction, resulting in abnormal results.

The measurement data of 2021-9-A is taken as an example to illustrate this situation, as shown in Figure S3 (a). The number of peak points of $SO_2$ in this set of measurement data is much higher than the number of peak points of $CO_2$. Therefore, a one-to-many matching result of peak points is bound to occur when using the DTW to match all peak points, which is the case at time points 110, 290, and 300. However, these peak trends do not allow us to directly select the most appropriate peak points. Therefore, it is necessary to ensure objective judgment conditions to eliminate unreasonable peak matching, so as to facilitate subsequent FSC and EF calculation.

The method adopted in this study was to perform DTW matching on 148 groups of collected data and summarize all matching results, totaling 911 matching pairs. This contains both the matches that can be used to calculate EF and the incorrect matches. The Manhattan distance of each pair was calculated separately and divided into two categories using the K-means algorithm and the threshold for classification was determined. If the value is smaller than this threshold, the matching pair is considered reasonable; otherwise, the matching pair is abnormal. When abnormal matching is identified, the matching result with the maximum concentration of $SO_2$ and $CO_2$ is selected as the global optimal peak point. This set of results is collected by the gas sensor at the position of the maximum gas concentration. Compared with other peak trends, it is less affected by various uncertainty factors and more conducive to improving the accuracy of the EF calculation. As seen in Figure 3S (b), several matching results at time points 290 and 300 were eliminated through the above process, and, finally, the matching result at time point 110 was selected as the global optimal peak point.

[Figure]

(a) DTW matching results                    (b) Global optimal peak points marked in black circle

**Figure S3. 2021-9-8A DTW matching results of measurement data and the final global optimal peak point.**

**5.The final step in the algorithm is a quality assessment of the derived pairs. Here, the authors apply 16 different index definitions on data sets created by "10000 self-development sampling" and then use a complex scheme to extract a high-level quality index from this set of results. I must admit that I did not follow this indexing in detail, as it appears completely arbitrary and pointless to me.**

We have added the following in the appendix to illustrate the effect of this step.

All experiments used to calculate the EF and FSC are based on the measurement of the ship plume at sea rather than the measurement experiment conducted in the laboratory. Therefore, various factors may have interfered with the obtained data. For a specific monitoring process, an objective and quantitative evaluation is needed to assess whether the calculation results are effective and feasible. Although many researchers have analyzed the uncertainty of the measurement process, qualitative analysis is the main method used in previous studies, while quantitative analysis is relatively lacking. Therefore, we put forward the concept of assessing data quality according to the actual application requirements and designed a calculation method based on the principle of self-expanding sampling through quantified score values to represent data quality. To further illustrate the effectiveness of the evaluation method for measurement data quality proposed in this paper, more measurement data are listed here and shown in Figure S4.

[Figure]

**Figure S4. Six typical data sets**

The presented six data sets vary in quality. Among them, the peak trends of the 2021-3-25C and 2021-3-26J data groups are very clear and reasonable matching relationships can be established for each peak in the $SO_2$ and $CO_2$ sequences; thus, the data quality of these two groups is good. In 2020-12-25A and 2021-1-22A, there are few time points at which the peak height of $SO_2$ and $CO_2$ is consistent, but for a single sequence the peak trend is clear and a more suitable global optimal peak can be selected from it. For example, the peak combination at time point 300 can be selected as the global optimal peak value in 2020-12-25A, and the peak combination at time point 850 can be selected as the global optimal peak value in 2021-1-22A. Thus, the data quality of these two groups is average. In both 2021-5-12A and 2021-5-19A, the $SO_2$ sequence showed sharp fluctuations, which resulted in a much higher number of peaks than in the $CO_2$ sequence. In addition, the peak height of the $SO_2$ sequence was almost maintained at the same level and it was difficult to establish a reasonable matching relationship between them; thus, the data quality of these two groups is poor.

The results of the above six data groups evaluated using the quality evaluation method presented in this paper are shown in Table S3. The joint evaluation results of 2021-3-25C and 2021-3-26J are both close to 0, indicating good-quality data. The $SO_2$ and $CO_2$ sequences of 2021-3-25C are highly synchronous, and the peak trends are similar. Although the two groups of peak trends of 2021-3-26J can establish a good matching relationship overall, there is a certain delay in the timing of the $SO_2$ and $CO_2$ sequences, and the synchronization degree is lower than that of 2021-3-25C. As such, the joint evaluation result of 2021-3-26J is slightly higher than that of 2021-3-25C. The joint evaluation results of 2020-12-25A and 2021-1-22A are both around 0.5, indicating medium-quality data. The peak heights of the $SO_2$ and $CO_2$ sequences in 2020-12-25A differ significantly more than the peak heights of the two sequences in 2021-1-22A. Therefore, the joint evaluation result of 2021-1-22A is smaller, with 0.5102. The result of each

evaluation index of 2021-5-12A and 2021-5-19A is 1, thus the joint evaluation result is also 1, indicating poor-quality data.

In summary, by comparing the joint evaluation results of six data groups of different quality, the feasibility of the proposed measurement data quality evaluation method could be verified.

**Table S3. Algorithm screening of six measurement data groups (2021-3-25C, 2021-3-26J, 2020-12-25A, 2021-1-22A, 2021-5-12A, and 2021-5-19A). The specific values and evaluation results of information entropy ($SO_2$), sample entropy ($SO_2$), skewness ($CO_2$), quartile spacing ($SO_2$), and the results of the joint evaluation of the four indexes are presented.**

| Data ID | Algorithm | Information entropy | | Sample entropy | | Skewness | | Quartile spacing | | Result of joint evaluation |
|---|---|---|---|---|---|---|---|---|---|---|
| | | Value | EVAL | Value | EVAL | Value | EVAL | Value | EVAL | |
| **2021-3-25C** | √ | 5.0189 | 1 | 0.0578 | 1 | 1.2260 | 0.1070 | 0.3307 | 1 | 0.1070 |
| **2021-3-26J** | √ | 3.5578 | 0.3117 | 0.0316 | 1 | 1.2653 | 0.0698 | 0.2871 | 1 | 0.1908 |
| **2021-5-12A** | √ | 4.2887 | 1 | 0.7358 | 1 | 0.5386 | 1 | 0.0811 | 1 | 1 |
| **2021-5-19A** | × | 4.0232 | 1 | 1.7174 | 1 | 1.7709 | 1 | 0.1481 | 1 | 1 |
| **2020-12-25A** | √ | 3.8523 | 1 | 0.1345 | 0.5220 | 1.2054 | 0.2001 | 0.1542 | 0.9272 | 0.5498 |
| **2021-1-22A** | √ | 3.3620 | 0.5102 | 0.0257 | 1 | 0.8295 | 1 | 0.2573 | 1 | 0.5102 |

**"×" indicates that the screening algorithm is inconsistent with subjective judgment**

**6.In summary, I think the manuscript tries to solve the wrong problem with a complex and, in many respects, arbitrary algorithm. Instead, the authors should follow a simple and physics-based approach by comparing the SO2 and CO2 values integrated over the plume transect after allowing for a time shift correcting for the different response times of the two instruments used.**

I believed that the significance of this research can be explained to a more extent through the supplementary notes in the above appendix. At the same time, we also optimize some arguments in the paper for clearer expression. In addition, we also added a comparative experiment in the appendix to illustrate the significance of the research.

In studies by Zhou et al. (2019, 2020), a total of 34 groups of "$SO_2$+$CO_2$" plume measurement data and actual FSC values obtained through ship boarding and extraction of oil samples were included, and an experience-based FSC calculation method was proposed. In a study by Beecken et al. (2015), an empirical calculation method based on the "area ratio" was proposed. A method based on the filtering principle was proposed in Krause et al. (2023). To verify the effect of the calculation method outlined in this study, data from Zhou et al. (2019, 2020) were used for comparison. The method in this paper and the above methods were used to calculate the FSC and compare the results with their actual value. Through comparison, it can be seen that the accuracy obtained using the method outlined in this paper is comparable to that of the method proposed by Zhou et al. (2019, 2020), and higher than that obtained through other methods. However, the method presented here is an objective and quantitative one rather than an experience-based approach.

**Table S4. Data from Zhou et al. (2019, 2020) were used to calculate the FSC through different methods. The deviation of the actual value from the estimated value is given in parentheses.**

| Plume ID | True | The | Proposed | Zhou et al., | The "area ratio" | Filter | based |
|---|---|---|---|---|---|---|---|

| | FSC | Method | (2019, 2020) | method | method |
|---|---|---|---|---|---|
| 2018-8-6A | 0.106 | 0.17(0.064) | 0.105(-0.001) | 0.298(0.192) | 0.603(0.497) |
| 2018-8-6B | 0.131 | 0.53(0.399) | 0.126(-0.005) | 0.276(0.145) | 0.264(0.133) |
| 2018-8-7A | 0.095 | 0.828(0.733) | 0.083(-0.012) | 0.297(0.202) | 0.17(0.075) |
| 2018-8-7B | 0.192 | 0.303(0.111) | 0.195(0.003) | 0.221(0.029) | 0.22(0.028) |
| 2018-8-9A | 0.04 | 0.118(0.078) | 0.038(-0.002) | 0.062(0.022) | 0.078(0.038) |
| 2018-8-9B | 0.211 | 0.171(-0.04) | 0.216(0.005) | 0.143(-0.068) | 0.185(-0.026) |
| 2018-8-28A | 0.079 | 0.155(0.076) | 0.085(0.006) | 0.207(0.128) | 0.173(0.094) |
| 2018-8-28B | 0.087 | 0.147(0.06) | 0.086(-0.001) | 0.125(0.038) | 0.145(0.058) |
| 2018-10-30A | 0.069 | 0.469(0.4) | 0.054(-0.015) | 0.07(0.001) | 0.129(0.06) |
| 2018-11-13A | 0.239 | 0.216(-0.023) | 0.226(-0.013) | 0.283(0.044) | 0.757(0.518) |
| 2018-11-26A | 0.069 | 0.034(-0.035) | 0.045(-0.024) | 0.026(-0.043) | 0.023(-0.046) |
| 2018-11-29A | 0.103 | 0.059(-0.044) | 0.068(-0.035) | 0.034(-0.069) | 0.035(-0.068) |
| 2018-12-18A | 0.037 | 0.024(-0.013) | 0.026(-0.011) | 0.016(-0.021) | 0.018(-0.019) |
| 2018-12-19A | 0.043 | 0.628(0.585) | 0.026(-0.017) | 0.002(-0.041) | 4.48(4.437) |
| 2019-1-3A | 0.369 | 0.226(-0.143) | 0.37(0.001) | 0.265(-0.104) | 0.372(0.003) |
| 2019-1-21A | 0.059 | 0.026(-0.033) | 0.027(-0.032) | 0.017(-0.042) | 0.006(-0.053) |
| 2019-1-22A | 0.104 | 0.056(-0.048) | 0.067(-0.037) | 0.059(-0.045) | 0.063(-0.041) |
| 2019-1-24A | 0.074 | 0.066(-0.008) | 0.097(0.023) | 0.046(-0.028) | 0.058(-0.016) |
| 2019-3-18A | 0.222 | 0.189(-0.033) | 0.225(0.003) | 0.152(-0.07) | 0.215(-0.007) |
| 2019-3-18B | 0.222 | 0.249(0.027) | 0.207(-0.015) | 0.151(-0.071) | 0.175(-0.047) |
| 2019-3-22A | 0.099 | 0.056(-0.043) | 0.077(-0.022) | 0.032(-0.067) | 0.044(-0.055) |
| 2019-3-22B | 0.099 | 0.061(-0.038) | 0.062(-0.037) | 0.046(-0.053) | 0.08(-0.019) |
| 2019-3-22C | 0.042 | 0.047(0.005) | 0.046(0.004) | 0.033(-0.009) | 0.058(0.016) |
| 2019-3-29A | 0.035 | 0.051(0.016) | 0.051(0.016) | 0.031(-0.004) | 0.068(0.033) |
| 2019-4-1A | 0.079 | 0.045(-0.034) | 0.065(-0.014) | 0.027(-0.052) | 0.079(0) |
| 2019-4-1B | 0.079 | 0.064(-0.015) | 0.064(-0.015) | 0.049(-0.03) | 0.039(-0.04) |
| 2019-4-3A | 0.013 | <0.02(N) | <0.02(N) | 0(-0.013) | 0.003(-0.01) |
| 2019-4-3B | 0.092 | 0.053(-0.039) | 0.052(-0.04) | 0.035(-0.057) | 0.057(-0.035) |
| 2019-4-12A | 0.004 | <0.02(N) | <0.02(N) | 0(-0.004) | 0(-0.004) |
| 2019-4-12B | 0.08 | 0.078(-0.002) | 0.08(0) | 0.054(-0.026) | 0.111(0.031) |
| 2019-4-15A | 0.044 | 0.035(-0.009) | 0.035(-0.009) | 0.006(-0.038) | 0.122(0.078) |
| 2019-4-15B | 0.168 | 0.173(0.005) | 0.162(-0.006) | 0.118(-0.05) | 0.137(-0.031) |
| 2019-4-15C | 0.044 | 0.035(-0.009) | 0.035(-0.009) | 0.006(-0.038) | 0.122(0.078) |
| 2019-4-15D | 0.168 | 0.148(-0.02) | 0.154(-0.014) | 0.133(-0.035) | 0.116(-0.052) |